# Is Male Sex A Prognostic Factor in Papillary Thyroid Cancer?

**DOI:** 10.3390/jcm10112438

**Published:** 2021-05-31

**Authors:** Aleksandra Gajowiec, Anna Chromik, Kinga Furga, Alicja Skuza, Danuta Gąsior-Perczak, Agnieszka Walczyk, Iwona Pałyga, Tomasz Trybek, Estera Mikina, Monika Szymonek, Klaudia Gadawska-Juszczyk, Artur Kuchareczko, Agnieszka Suligowska, Jarosław Jaskulski, Paweł Orłowski, Magdalena Chrapek, Stanisław Góźdź, Aldona Kowalska

**Affiliations:** 1ESKULAP Student Scientific Organization, Collegium Medicum, Jan Kochanowski University, 25-317 Kielce, Poland; ania.chromik@op.pl (A.C.); kiniafurga@gmail.com (K.F.); alicja.skuzaf@gmail.com (A.S.); 2Collegium Medicum, Jan Kochanowski University, 25-317 Kielce, Poland; danutagp@o2.pl (D.G.-P.); a.walczyk@post.pl (A.W.); iwonapa@tlen.pl (I.P.); jaroslawja@poczta.fm (J.J.); pawelor@interia.pl (P.O.); stanislawgozdz1@gmail.com (S.G.); aldonako@onkol.kielce.pl (A.K.); 3Endocrinology Clinic, Holycross Cancer Centre, S. Artwińskiego Str. 3, 25-734 Kielce, Poland; trytom1@tlen.pl (T.T.); esterami@tlen.pl (E.M.); christ76@interia.pl (M.S.); klaudiagdwsk@op.pl (K.G.-J.); arturkuchareczko@gmail.com (A.K.); a.suligowska@wp.pl (A.S.); 4Faculty of Natural Sciences, Jan Kochanowski University, 25-406 Kielce, Poland; Magdalena.Chrapek@ujk.edu.pl; 5Clinical Oncology, Holycross Cancer Centre, S. Artwińskiego Str. 3, 25-734 Kielce, Poland

**Keywords:** papillary thyroid cancer, male sex, response to therapy, risk stratification

## Abstract

Identifying risk factors is crucial for predicting papillary thyroid cancer (PTC) with severe course, which causes a clinical problem. The purpose of this study was to assess whether male sex can be such a predictive factor and to verify whether including it as a predictive factor of high initial risk of recurrence/persistence would help to enhance the value of the American Thyroid Association initial risk stratification system (ATA). We retrospectively analyzed 1547 PTC patients (1358 females and 189 males), treated from 1986 to 2018. The relationship between sex and clinicopathological features, response to therapy, and disease status was assessed. Men with PTC showed some adverse clinicopathological features more often than women, including angioinvasion, lymph node metastases, and tumor size > 40 mm. There were sex-related disparities with respect to response to initial therapy and final follow-up. Male sex is associated with some unfavorable clinicopathological features of PTC, which may affect response to initial therapy or final disease status. In our study, modification of the ATA system by including male sex as a risk factor does not enhance its value. Thus, further studies are needed to assess whether males require treatment modalities or oncological follow-up protocols that are different from those of females.

## 1. Introduction

Thyroid carcinoma (TC) is the most common malignancy of the endocrine system. According to the World Health Organization database, 567,233 new cases of TC were reported in 2018: 130,889 cases in men and 436,344 in women [1]. Exposure to radiation, low iodine intake, genetics, obesity, and smoking are associated with an increased risk of developing TC [2,3]. Well-differentiated TC (DTC), which originates from thyroid follicular epithelial cells, is the most common type of TC, accounting for approximately 90% of cases [4]. The prevalence of DTC, particularly papillary thyroid cancer, continues to rise worldwide. 

The sharp increase in the incidence of TC (mostly small cancers) over the past three decades is mainly due to the development of widely available and sensitive detection methods such as thyroid ultrasound and to increased availability of ultrasound-guided fine-needle aspiration [4]. Furthermore, we observe an increase in the incidence of larger thyroid cancers due to exposure to radiation, low iodine intake, genetics, obesity, and smoking [5,6]. PTC, the most common type of DTC, is a relatively benign tumor with a favorable prognosis. However, there is a disparity in morbidity between women and men. Previous studies report that the incidence of thyroid cancer is 3–5-fold higher in women [7,8]; however, reports suggest that the disease is more aggressive in men [9]. Male sex is associated with a worse prognosis for PTC, especially more advanced disease, and aggressive subtypes [9,10]. Men tend to be diagnosed at an older age and have shorter disease-free survival [10,11]. In females, the age-specific incidence rate rises sharply at the beginning of the reproductive years, peaking at 40–49 years, while in men, the peak is at 60–69 years. The incidence rates equalize by 85 years. However, the link between sex and presentation of DTC is ambiguous. 

The aim of the present study was to analyze the effect of male sex on occurrence of PTC with unfavorable clinicopathological features and to examine whether men show a different response to therapy. Another objective was to verify whether male sex should be considered as a risk factor for recurrent/persistent disease according to the American Thyroid Association (ATA) initial risk stratification system which would help to enhance the accuracy of this system. 

## 2. Experimental Section

The medical records of 1900 Caucasian patients with PTC, all of whom underwent total thyroidectomy or lobectomy during 1986–2018, were examined retrospectively. The following data were obtained: type of cancer, sex, prognostic clinicopathological features (i.e., age at diagnosis, tumor diameter, multifocality, lymph node metastases, angioinvasion, extrathyroidal extension, and distant metastases), response to initial therapy, and clinical outcome (i.e., no evidence of disease (NED), recurrence, persistent disease, or death). Patients with follow-up less than 12 months, lacking complete data, and those with a non-invasive follicular thyroid neoplasm with papillary-like nuclear features (NIFTP) were excluded. Patients who were lost to follow-up for reasons other than death were also excluded. The data of 25 patients who underwent total thyroidectomy between 1986 and 1999 were obtained from other centers. The study ultimately included 1547 patients. 

All patients included in the study underwent surgery as the primary treatment: lobectomy, total thyroidectomy, or total thyroidectomy with central compartment lymphadenectomy. The extent of the surgery and surgical procedures has been described by us previously [12]. 

Postoperative TNM staging of all included patients was re-classified according to the most recent 8th edition of the American Joint Committee on Cancer (AJCC)/Union for International Cancer Control (UICC) TNM staging system and the American Thyroid Association (ATA) modified initial risk stratification system [13] 

Response to therapy was assessed 1 year after initial treatment. Patients treated with radioiodine (131I) were assessed for their response to initial therapy (surgery plus 131I) using the criteria proposed by Tuttle et al. [14], which were accepted by the ATA [15]. The procedures performed during the follow-up and assessment of treatment responses have been described previously [16]. 

The response to initial therapy (either total thyroidectomy or lobectomy) of patients not treated with radioiodine was assessed using criteria proposed by Momesso and Tuttle [17]. The procedures used to monitor disease course from the end of surgical treatment to evaluation of response were described previously [18]. Treatment response was classified as excellent, indeterminate, biochemically incomplete, or structurally incomplete.

Information about recurrence was obtained from medical records. Recurrence was defined as biochemical or structural disease after a period of NED (at least 12 months after initial therapy). 

The clinical status of the patient was assessed based on medical records and by applying the latest ATA guidelines. Patients were then assigned to one of four groups: NED, persistent disease, death from cancer, and death from other causes. The follow-up results were finally concluded on 31 May 2020.

The study plan was accepted by the Bioethics Committee at Jan Kochanowski University in Kielce, Poland (Number-7/2020). Written informed consent was not required because the data were obtained retrospectively.

Continuous data are expressed as the median, lower (Q1) and upper (Q3) quartiles, and range (minimum–maximum). Categorical data are expressed as numbers and percentages. Group comparisons were performed using the Chi-squared or Fisher’s exact test (categorical variables), the t-test (continuous, normally distributed variables) or the Mann–Whitney test (continuous, non-normally distributed variables). The normality of data was checked using the Shapiro–Wilk test. 

Univariate and multivariate logistic regression models were used to analyze excellent responses to initial therapy and NED during follow-up; in these models, odds ratios (ORs) with 95% confidence intervals (95% CIs) were calculated. Statistical tests were two-tailed, and a *p*-value of <0.05 was considered significant. All statistical analyses were performed using the R software package version 3.6.1 (R Core Team (2019). R: A language and environment for statistical computing. R Foundation for Statistical Computing, Vienna, Austria. URL https://www.R-project.org/, accessed on 13 May 2021).

## 3. Results

### 3.1. Characteristics of the Study Group

The database included 1547 cases (1358 females (87.8%) and 189 males (12.2%); median age at diagnosis, 50 years; range, 15–85 years). Of these, 64.0% (990/1547) were younger than 55 years. The most frequent cancer histological variant was classic variant (71.8%; 1111/1547). Angioinvasion was present in 4.7% of patients (72/1547) and multifocality in 26.1% (404/1547). In 3.0% (46/1547) of cases, tumor diameter was >40 mm (median tumor size, 8.0 mm; range, 0.3–110.0 mm). Extrathyroidal extension (ETE) occurred in 19.2% of cases: minor ETE in 17.6% (272/1547) and gross ETE in 1.6% (25/1547). Lymph node metastases were identified in 14.5% of cases (225/1547); 1.0% (15/1547) of patients developed distant metastases (M1). Only 5.3% (81/1547) of cases had disease stage >I according to the AJCC TNM Staging System for Differentiated and Anaplastic Thyroid Cancer (8th Edition) criteria. The ATA initial risk stratification system rated the majority (68.2%, 1055/1547) as low risk of recurrence/persistence. Radioactive iodine therapy was applied in 1054/1547 (68.1%) patients. The characteristics of the study group are presented in Table 1.

### 3.2. Response to Initial Therapy, Recurrence, and Final Clinical Status 

#### 3.2.1. Response to Initial Therapy

Response to treatment was assessed in all 1547 patients. The following results were obtained: excellent response to initial therapy was observed in 1259 patients (81.4%), and an indeterminate response was observed in 234 (15.1%). Twenty-two patients (1.4%) achieved a biochemically incomplete response, and 32 (2.1%) achieved a structurally incomplete response. 

#### 3.2.2. Recurrence

The rate of recurrence after an excellent response to initial therapy was 19/1259 (1.5%). 

#### 3.2.3. Final Clinical Status

At final follow-up, NED was found in 1423/1547 cases (92.0%). Non-NED was observed in 124/1547 (8.0%) patients (indeterminate, 102 (6.6%); biochemically persistent, 12 (0.8%); and structurally persistent, 10 (0.6%)). The median follow-up time was 6.0 years (range, 1.0–34.0 years). Overall, sixteen patients died (16/1547; 1.0%). There were no cancer-related deaths. The results are presented in Table 2. 

### 3.3. Baseline Characteristics According to Sex

A comparison of baseline tumor characteristics between the sexes is presented in Table 3. The research group comprised 1358 women and 189 men. In this study, PTC was more than seven times as common in women as in men. The median age was 50 years (range, 15–85) for women and 51 years (range, 20–80) for men. Median tumor size in patients did not differ according to sex and was 8.0 mm (range, 0.3–80.0 mm for women and 0.3–110.0 mm for men). Significantly fewer females had tumors > 40 mm: 2.4% (32/1358) vs. 7.4% (14/189); *p* = 0.0001). The most frequent papillary cancer histologic variant in both sexes was classic (71.1% of women, 76.7% of men). The incidence of angioinvasion differed according to sex: 7.9% (15/189) of males and 4.2% (57/1358) of females (*p* = 0.0222). There was a significant difference in occurrence of regional lymph node metastases in males and females (Table 3). The TNM classification system classified nodal metastases as 1a or 1b in 23.8% (45/189) of males and in 13.2% (180/1358) of females (*p* < 0.0001). There was a significant sex-related difference in disease stage, as assessed by the AJCC TNM Staging System for Differentiated and Anaplastic Thyroid Cancer (8th Edition) criteria. Men were more likely to have disease stage > I. Only 0.4% (5/1358) of women (compared with 1.6% (3/189) of men) were diagnosed with the most advanced stage of cancer (IVa and IVb; *p* = 0.0005). According to the ATA initial risk of recurrence and persistence stratification system significantly more men (7.9%, 15/189) than women (3.7%, 50/1358) were in the high-risk group (*p* = 0.0054). Among women, 67.4% (915/1358) were treated with radioiodine, while among men 73.5% (139/189); significantly more men than women received 131I treatment more than once (10.8% of women, 15.9% of men, *p* = 0.0377). 

### 3.4. Response to Initial Therapy, Recurrence, and Final Clinical Status According to Sex 

#### 3.4.1. Response to Initial Therapy

Univariate analysis identified significant differences between the sexes with respect to excellent and non-excellent responses to initial therapy. Results were confirmed by multivariate analysis, meaning that male sex is an independent risk factor for a worse response to therapy. Other risk factors such as angioinvasion and nodal metastases may affect the results of univariate analysis. In patients with PTC, 82.8% (1125/1358) of females versus 70.9% (134/189) of males showed an excellent response to initial therapy. In this group, a non-excellent response was observed in 17.2% (233/1358) of females and 29.1% (55/189) of males (*p* < 0.0001). The results are presented in Table 4.

Univariate and multivariate analyses were performed to ascertain which clinicopathological features may affect achievement of a non-excellent response to treatment. The results are presented in Table 5.

In the study group, univariate analysis identified male sex as the single factor that increased the probability of a non-excellent response to therapy, with an OR of 1.98 (95% CI, 1.4–2.8, *p* = 0.0001); this was confirmed by multivariate analysis (OR 1.62; 95% CI, 1.12–2.35; *p* = 0.0109). 

Other significant predictors that may increase the chances of achieving a response other than excellent were primary tumor size >40 mm, multifocality, nodal metastases, angioinvasion, and extrathyroidal (both minor and gross) extension. Multivariate analysis confirmed the effects of all these factors. According to univariate analysis, being in an intermediate and high-risk group also increased the chances of a non-excellent response to therapy. The “ATA initial risk” variable, along with other clinicopathological parameters, was not entered into the multivariable logistic regression model to avoid introduction of multicollinearity. Age ≥55 at diagnosis in this case was neither a positive nor a negative predictor, with an OR of 0.83 (95% CI, 0.63–1.09; *p* = 0.1875). Presence of distant metastases was not entered in the analysis as a predictor because no patient with distant metastases achieved excellent response.

#### 3.4.2. Recurrence

Multivariate analysis could not have been conducted due to the fact that cancer recurrences were observed only in females (19 patients). Result of such analysis would not be reliable. 

#### 3.4.3. Final Clinical Status

At the final follow-up, 93.1% (1264/1358) of women and 84.1% (159/189) of men were classified as having NED (Table 6). We performed univariate and multivariate logistic regression analyses to ascertain which clinicopathological features affect achievement of a non-NED diagnosis at the final follow-up. The results are presented in Table 7. Univariate analysis identified male sex as a factor that increased the probability of non-NED at the final clinical assessment. This was confirmed by multivariate analysis, with an OR of 1.96 (95% CI, 1.21–3.17; *p* = 0.0062). Univariate and multivariable analyses identified nodal metastases and minor and gross ETE as predictors with a significant effect on persistence of disease. The “ATA initial risk” variable, along with other clinicopathological parameters, was not entered into the multivariable logistic regression model to avoid introduction of multicollinearity.

As a summary for this research, we made an attempt to modify the ATA stratification system. Male sex was added to the prognosis worsening factors that qualify for non-low risk. Then, the sensitivity and specificity of modified and unmodified systems were compared. Such modification improves sensitivity of the stratification system (68.8% vs. 62.5%, *p* < 0.0001) but also significantly worsens its specificity (75.2% vs. 67.6%, *p* < 0.0001). Data are presented in Table 8. 

## 4. Discussion

The results of this study show that males with PTC are more likely to present with some unfavorable clinicopathological risk factors such as lymph node metastases and angioinvasion. Liu et al. made similar observations [19]. However, we classified nodal metastases as N1 in a smaller percentage of patients than Liu et al.: 23.8% of males and 13.2% females versus 29.1% and 19.3%, respectively. Moreover, Liu et al. found that men are more often diagnosed with a larger primary tumor [19]: mean tumor size (22.48 ± 21.79 for men vs. 16.93 ± 17.75 mm for women). However, we found this not to be the case (median tumor size: 8.0 mm for women and 8.0 mm for men in our study). Over 80% of our patients (85.9%) were diagnosed with a primary tumor smaller than 20 mm, although a tumor diameter larger than 40 mm was significantly more common in men. 

Our findings indicate no differences between the sexes with respect to age at diagnosis. This disagrees with the results of two large studies based on U.S. and Chinese populations [19,20]. The median age of male and female patients was very similar between these studies: age at diagnosis was 49 years for women and 53 years for men. In our study, the median age for women was 50 years, and that for men was 51 years. 

We also found that angioinvasion was significantly more common in male patients. However, its effect, particularly in males has not been described yet. Angioinvasion is thought to predict an increased risk of recurrence and poor prognosis, with treatment implications [21]. What is more, in accordance with the ATA Management Guidelines criteria, invasion of blood vessels is a crucial feature that has to be absent in the low-risk PTC. 

The aim of the present study was to evaluate whether male sex is associated with worse course and prognosis of PTC. By the term “worse prognosis”, we mean a response to initial therapy that is other than excellent; this is because PTC-related death, although not frequent, usually affects patients with an incomplete response to initial treatment. This is in agreement with the study by Sapuppo et al. [22], who stated that patients not cured after initial therapy are at increased risk of persistent disease and a worse outcome. We identified minor extrathyroidal extensions, multifocality, angioinvasion, and distant and nodal metastases as factors that significantly increase the risk of a non-excellent response to treatment. This was confirmed by multivariate analysis.

Therefore, we assessed the effect of male sex on achieving a non-excellent response to initial therapy. Univariate analysis identified male sex as being associated with a non-excellent response to initial treatment, which was confirmed by multivariate analysis. Mendoza et al. identified male sex, along with the extent of the disease, multifocality, and lymph node metastasis, as factors that increase the risk of an incomplete response [23]. Nevertheless, multivariate analysis in the cited study did not include distant metastasis and angioinvasion. In our study, angioinvasion was a significantly relevant factor contributing to non-excellent response (OR, 1.83) [23]. The Sapuppo study indicated that male sex has an impact on persistent disease status (OR, 1.7), but they did not identify distant metastasis and angioinvasion as contributing factors, which might be the reason for the inconsistencies in the results [22].

By “worse course”, we mean the presence of persistent disease (described in this study as non-NED at the final follow-up). Multivariable analysis suggested that patients diagnosed with extrathyroidal extension (both minor and gross) and nodal metastases were at risk of non-NED at the final clinical assessment. 

Univariate analysis of PTC patients identified male sex as a factor that may increase the probability of a non-NED diagnosis at the final clinical assessment. Such dependency was confirmed by multivariate analysis. 

We also examined the frequency of disease recurrence. Recurrence was defined as the appearance of disease visualized in imaging studies and/or an increase in expression of tumor markers after a period of NED (as defined by ATA). It was observed in nineteen patients with PTC.

Chow [24] and Toniato [25] found that the risk of recurrence was increased by many factors, including age at diagnosis, tumor size, and nodal and distant metastases. In both cited studies, the follow-up period was longer than that in our study: Toniato et al. [25] used an average follow-up period of 7.8 years, whereas that used by Chow et al. [24] was 8.4 years. The frequency of recurrence may differ between studies because they use different definitions. For example, in a multicenter study by Zahedi et al. each of the co-authors defined recurrence of disease differently and did not specify the disease-free term. Hence, some patients with persistent disease were included in the recurrence group [26]. In our study, recurrence was defined as a relapse of disease in patients who achieved an excellent response to therapy after the NED period. It was a very rare phenomenon, seen only in nineteen of our patients. It was not possible to assess statistical significance of sex impact on recurrence because in our cohort cancer recurrences were observed only in females.

We also reported no PTC-related deaths; therefore, we could not examine the effect of sex on death.

Analysis of other studies conducted in European and American populations indicated that male sex is not associated with higher mortality (cancer-related death) [6,20,25,27,28].

Some adverse clinicopathological features (assessed immediately after surgery) were more common in men. Sex affects the clinical final status of disease (NED vs. non-NED). This is in line with the findings of Tuttle et al., Kowalska et al., Momesso et al. and Castagna et al. [14,16,29,30], who showed that the delayed risk stratification system based on response to initial therapy correlated more strongly with final status of disease (NED/non-NED) than the early stratification systems created by the ATA. In our study, male sex was a factor that predicted a worse response to initial therapy, and there was a relationship between sex and final status of disease.

Authors of many previous studies have attempted to verify that the current stratification system is reliable and effectively predicts the prognosis for patients after initial DTC treatment [14,31,32]. In some studies, value of new markers for stratification was examined. For example, Vuong et al. and Moon et al. found the correlation between concomitant BRAF and TERT promoter mutations with more aggressive disease course [33,34]. This finding was included in Clinical Practice Guidelines by the European Society for Medical Oncology and added as a high risk of persistent/recurrent disease predictor in PTC patients [35].

In our research, we tried to modify the ATA stratification system by adding male sex to the high-risk predictors. Although such modification improves the sensitivity of the test (68.8% vs. 62.5%), it does not enhance its specificity. Further studies on larger groups of patients should be carried out to assess whether males require treatment modalities or oncological follow-up protocols that are different from those of females as our results in this matter are inconclusive.

Our study has some limitations. First, the number of patients with advanced clinical disease was small, although almost all patients with PTC treated in our center from 2000 to 2018 were included. Second, the follow-up period was short. Follow-up for more than 6 years may reveal more information about recurrence or death related to PTC.

## 5. Conclusions

Male sex is associated with some unfavorable clinicopathological characteristics of PTC and contributes to a worse response to initial therapy or final disease status. Thus, further studies on larger groups of patients should be carried out to assess whether males require treatment modalities or oncological follow-up protocols that are different from those of females.

## Figures and Tables

**Table 1 jcm-10-02438-t001:** Characteristics of the study group.

	Total (*n* = 1547)
Sex, *n* (%)	
Female	1358 (87.8%)
Male	189 (12.2%)
Median age, years (Q1–Q3; range)	50.0 (39.0, 59.0; 15.0–85.0)
Age at diagnosis, ≥55 years, *n* (%)	557 (36.0%)
Median tumor size, mm (Q1–Q3; range)	8.0 (5.0–14.0; 0.3–110.0)
Tumor diameter, >40 mm, *n* (%)	46 (3.0%)
Papillary cancer histologic variant, *n* (%)	
Classic	1111 (71.8%)
Follicular	363 (23.5%)
Oxyphilic	13 (0.8%)
Diffuse sclerosing	9 (0.6%)
Tall cell	4 (0.3%)
Other	47 (3.0%)
Multifocality, *n* (%)	
No	1143 (73.9%)
Yes	404 (26.1%)
Extrathyroidal extension, *n* (%)	
No	1250 (80.8%)
Minor	272 (17.6%)
Gross	25 (1.6%)
Angioinvasion, *n* (%)	
No	1475 (95.3%)
Yes	72 (4.7%)
Tumor stage, *n* (%)	
pT1a	980 (63.3%)
pT1b	349 (22.6%)
pT2	152 (9.8%)
pT3a	42 (2.7%)
pT3b	17 (1.1%)
pT4a	6 (0.4%)
pT4b	1 (0.1%)
Nodal metastases, *n* (%)	
N0a	801 (51.8%)
N0b	521 (33.7%)
N1a	124 (8.0%)
N1b	101 (6.5%)
Distant metastases, *n* (%)	
M0	1532 (99.0%)
M1	15 (1.0%)
TNM stage, *n* (%)	
I	1466 (94.8%)
II	71 (4.6%)
III	2 (0.1%)
IVa	1 (0.1%)
IVb	7 (0.5%)
ATA initial risk, *n* (%)	
Low	1055 (68.2%)
Intermediate	427 (27.6%)
High	65 (4.2%)
Radioactive iodine therapy	
No	493 (31.9%)
Yes	1054 (68.1%)
Therapy 131I > 1	
No	1371 (88.6%)
Yes	176 (11.4%)

TNM, tumor node metastases; ATA, American Thyroid Association; N0a, one or more cytologically or histologically confirmed benign lymph node; N0b, no radiologic or clinical evidence of locoregional lymph node metastases; N1a-b, metastases to regional lymph nodes.

**Table 2 jcm-10-02438-t002:** Response to initial therapy, recurrence, and final clinical status.

**Response to Initial Therapy, *n* (%)**	**Total (*n* = 1547)**
Excellent	1259 (81.4%)
Indeterminate	234 (15.1%)
Biochemical incomplete	22 (1.4%)
Structural incomplete	32 (2.1%)
**Recurrence after an Excellent Response to Initial Therapy, *n* (%)**	**Total (*n* = 1259)**
No	1240 (98.5%)
Yes	19 (1.5%)
**Final Follow-Up, *n* (%)**	**Total (*n* = 1547)**
NED	1423 (92.0%)
Indeterminate	102 (6.6%)
Biochemically persistent disease	12 (0.8%)
Structurally persistent disease	10 (0.6%)
**Death, *n* (%)**	**Total (*n* = 1547)**
No	1531 (99.0%)
Yes (TC-unrelated)	16 (1.0%)
**Median Follow-Up Time, Years (Range)**	6.0 (1.0–34.0)

NED, no evidence of disease; TC, thyroid cancer.

**Table 3 jcm-10-02438-t003:** Baseline characteristics according to sex.

	Female (*n* = 1358)	Male (*n* = 189)	Total (*n* = 1547)	*p*-Value
Median age, years(Q1–Q3; range)	50.0 (39.0–58.8; 15–85)	51.0 (41.0–60.0; 20–80)	50.0 (39.0–59.0; 15–85)	0.1297
Age at diagnosis, ≥55 years, *n* (%)	481 (35.4%)	76 (40.2%)	557 (36.0%)	0.1985
Median tumor size, mm (Q1–Q3; range)	8.0 (5.0–13.0; 0.3–80.0)	8.0 (5.0–23.0; 0.3–110)	8.0 (5.0–14.0; 0.3–110)	0.0776
Tumor diameter, >40 mm, *n* (%)	32 (2.4%)	14 (7.4%)	46 (3.0%)	0.0001
**Classic**	966 (71.1%)	145 (76.7%)	1111 (71.8%)	
Follicular	325 (23.9%)	38 (20.1%)	363 (23.5%)	
Oxyphilic	11 (0.8%)	2 (1.1%)	13 (0.8%)	
Diffuse sclerosing	9 (0.7%)	0 (0.0%)	9 (0.6%)	
Tall cell	4 (0.3%)	0 (0.0%)	4 (0.3%)	
Other	43 (3.2%)	4 (2.1%)	47 (3.0%)	0.623
**Multifocality, *n* (%)**				
No	1010 (74.4%)	133 (70.4%)	1143 (73.9%)	
Yes	348 (25.6%)	56 (29.6%)	404 (26.1%)	0.2404
Extrathyroidal extension, *n* (%)				
No	1107 (81.5%)	143 (75.7%)	1250 (80.8%)	
Minor	230 (16.9%)	42 (22.2%)	272 (17.6%)	
Gross	21 (1.5%)	4 (2.1%)	25 (1.6%)	0.1403
Angioinvasion, *n* (%)				
No	1301 (95.8%)	174 (92.1%)	1475 (95.3%)	
Yes	57 (4.2%)	15 (7.9%)	72 (4.7%)	0.0222
Tumor stage, *n* (%)				
pT1a	875 (64.4%)	105 (55.6%)	980 (63.3%)	
pT1b	315 (23.2%)	34 (18.0%)	349 (22.6%)	
pT2	118 (8.7%)	34 (18.0%)	152 (9.8%)	
pT3a	29 (2.1%)	13 (6.9%)	42 (2.7%)	
pT3b	17 (1.3%)	0 (0.0%)	17 (1.1%)	
pT4a	4 (0.3%)	2 (1.1%)	6 (0.4%)	
pT4b	0 (0.0%)	1 (0.5%)	1 (0.1%)	<0.0001
Nodal metastases, *n* (%)				
N0a	718 (52.9%)	83 (43.9%)	801 (51.8%)	
N0b	460 (33.9%)	61 (32.3%)	521 (33.7%)	
N1a	106 (7.8%)	18 (9.5%)	124 (8.0%)	
N1b	74 (5.4%)	27 (14.3%)	101 (6.5%)	<0.0001
Distant metastases, *n* (%)				
M0	1346 (99.1%)	186 (98.4%)	1532 (99.0%)	
M1	12 (0.9%)	3 (1.6%)	15 (1.0%)	0.4146
TNM stage, *n* (%)				
I	1298 (95.6%)	168 (88.9%)	1466 (94.8%)	
II	54 (4.0%)	17 (9.0%)	71 (4.6%)	
III	1 (0.1%)	1 (0.5%)	2 (0.1%)	
IVa	0 (0.0%)	1 (0.5%)	1 (0.1%)	
IVb	5 (0.4%)	2 (1.1%)	7 (0.5%)	0.0005
ATA initial risk, *n* (%)				
Low	941 (69.3%)	114 (60.3%)	1055 (68.2%)	
Intermediate	367 (27.0%)	60 (31.7%)	427 (27.6%)	
High	50 (3.7%)	15 (7.9%)	65 (4.2%)	0.0054
Radioactive iodine therapy				
No	443 (32.6%)	50 (26.5%)	493 (31.9%)	
Yes	915 (67.4%)	139 (73.5%)	1054 (68.1%)	0.0883
Therapy 131I > 1				
No	1212 (89.2%)	159 (84.1%)	1371 (88.6%)	
Yes	146 (10.8%)	30 (15.9%)	176 (11.4%)	0.0377

TNM, tumor node metastases; ATA, American Thyroid Association; N0a, one or more cytologically or histologically confirmed benign lymph node; N0b, no radiologic or clinical evidence of locoregional lymph node metastases; N1a-b, metastases to regional lymph nodes.

**Table 4 jcm-10-02438-t004:** Initial response to therapy according to sex.

Response to Initial Therapy, *n* (%)	Total (*n* = 1547)	Female (*n* = 1358)	Male (*n* = 189)	*p*-Value
Excellent	1259 (81.4%)	1125 (82.8%)	134 (70.9%)	
Non-excellent	288 (18.6%)	233 (17.2%)	55 (29.1%)	<0.0001

**Table 5 jcm-10-02438-t005:** Effect of clinicopathological factors on non-excellent response to initial therapy.

Clinicopathological Factor		Univariable OR	95% CI	*p*-Value	Multivariable OR	95% CI	*p*-Value
Sex *	Female	Ref. level			Ref. level		
	Male	1.98	1.4–2.8	0.0001	1.62	1.12–2.35	0.0109
Age at diagnosis, ≥55 years	No	Ref. level					
	Yes	0.83	0.63–1.09	0.1875			
Tumor diameter, >40 mm *	No	Ref. level			Ref. level		
	Yes	3.88	2.14–7.04	<0.0001	2.07	1.06–4.06	0.0338
Multifocality *	No	Ref. level			Ref. level		
	Yes	1.88	1.43–2.47	<0.0001	1.56	1.16–2.08	0.0028
Extrathyroidalextension *	0. no	Ref. level			Ref. level		
	1. minor	2.74	2.03–3.69	<0.0001	1.9	1.38–2.63	0.0001
	2. gross	7.28	3.26–16.28	<0.0001	3.33	1.37–8.14	0.0082
Angioinvasion *	No	Ref. level			Ref. level		
	Yes	4.05	2.5–6.56	<0.0001	1.83	1.06–3.15	0.0292
Nodal metastases *	No	Ref. level			Ref. level		
	Yes	4.07	3.5–5.53	<0.0001	2.69	1.91–3.77	<0.0001
Distant metastases *	No	Ref. level					
	Yes	NA(0in cell)					
ATA initial risk	0. low	Ref. level					
	1. intermediate	3.97	2.98–5.28	<0.0001			
	2. high	22.9	12.84–40.83	<0.0001			

OR, odds ratio; CI, confidence interval; NA(0in cell), not calculable (zero in cell); ATA, American Thyroid Association. *, variables with an asterisk were entered in a multivariable model.

**Table 6 jcm-10-02438-t006:** Final follow-up results according to sex.

Final Follow-Up, *n* (%)	Total (*n* = 1547)	Female (*n* = 1358)	Male (*n* = 189)	*p*-Value
NED	1423 (93.3%)	1264 (93.1%)	159 (84.1%)	
Non-NED	124 (6.7%)	94 (6.1%)	30 (15.9%)	< 0.0001

NED, no evidence of disease.

**Table 7 jcm-10-02438-t007:** Effect of clinicopathological factors on non-NED in follow-up.

Clinicopathological Factor		Univariable OR	95% CI	*p*-Value	Multivariable OR	95% CI	*p*-Value
Sex *	Female	Ref. level			Ref. Level		
	Male	2.54	1.63–3.95	<0.0001	1.96	1.21–3.17	0.0062
Age at diagnosis, ≥55 years	No	Ref. level					
	Yes	1.05	0.72–1.54	0.7918			
Tumor diameter, >40 mm *	No	Ref. level			Ref. Level		
	Yes	3.86	1.91–7.81	0.0002	1.63	0.72–3.67	0.24
Multifocality	No	Ref. level			Ref. Level		
	Yes	1.69	1.15–2.48	0.0077	1.25	0.82–1.9	0.3036
Extrathyroidal extension *	0. no	Ref. level			Ref. Level		
	1. minor	2.82	1.88–4.24	<0.0001	1.72	1.1–2.71	0.0185
	2. gross	8.94	3.82–20.91	<0.0001	2.98	1.07–8.32	0.0366
Angioinvasion *	No	Ref. level			Ref. Level		
	Yes	4.68	2.67–8.19	<0.0001	1.66	0.87–3.19	0.1272
Nodal metastases *	No	Ref. level			Ref. Level		
	Yes	6.11	4.14–9.01	<0.0001	3.89	2.5–6.03	<0.0001
Distant metastases *	No	Ref. level			Ref. Level		
	Yes	7.99	2.88–22.83	0.0001	3.18	0.93–10.94	0.0659
ATA initial risk	0. low	Ref. level					
	1. intermediate	4.76	3.12–7.27	<0.0001			
	2. high	16.11	8.83–29.38	<0.0001			

NED, no evidence of disease; OR, odds ratio; CI, confidence interval; ATA, American Thyroid Association. * variables with an asterisk were entered in a multivariable model.

**Table 8 jcm-10-02438-t008:** Comparison of sensitivity and specificity between modified and unmodified ATA systems.

	Unmodified ATA System	Modified ATA System	*p*-Value
Sensitivity	62.5%	68.8%	<0.0001
Specificity	75.2%	67.6%	<0.0001

ATA, American Thyroid Association.

## Data Availability

The data presented in this study are available on request from the corresponding author. The data are not publicly available due to their containing information that could compromise the privacy of research participants.

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
