# Peer review of "Is Male Sex A Prognostic Factor in Papillary Thyroid Cancer?"

_jcm, 2021, doi:10.3390/jcm10112438_

Round 1
Reviewer 1 Report
The manuscript is much improved and details the relevant factors in more detail. I have the following comments. 1. The abstract has a somewhat confusing conclusion regarding whether as it states that men are at higher initial risk of recurrence/persistence yet “male sex should not be included into the ATA stratification system as a risk factor for persistence/recurrence, because it does not enhance the value of this system.” This seems to be contradictory and also the results of this study are insufficient to warrant a change in the national stratification system. The conclusion in the manuscript seems to be much more appropriate than the conclusion in the abstract. Also, the results and discussion state that the study did not show a higher risk of recurrence in men. In fact, recurrence was only seen in women. This contradicts the statement in the abstract. 2. The introduction implies that the sharp rise in thyroid cancer is due to improved imaging and biopsy only. These are not the only causes of increase in thyroid cancer. There are likely multiple other factors involved. For example, more detailed pathology techniques may account for many of the small cancers identified.Author Response
Comment 1:
“The manuscript is much improved and details the relevant factors in more detail. I have the following comments. The abstract has a somewhat confusing conclusion regarding whether as it states that men are at higher initial risk of recurrence/persistence yet “male sex should not be included into the ATA stratification system as a risk factor for persistence/recurrence, because it does not enhance the value of this system.” This seems to be contradictory and also the results of this study are insufficient to warrant a change in the national stratification system. The conclusion in the manuscript seems to be much more appropriate than the conclusion in the abstract.”
We are grateful for the review of our manuscript. The clarification has been added in the abstract (Page 1, Line 34-37). Modification of ATA stratification system does not enhance its value, but further studies should be carried out to assess whether males require different treatment modalities or oncological follow-up protocols.
Comment 2:
“Also, the results and discussion state that the study did not show a higher risk of recurrence in men. In fact, recurrence was only seen in women. This contradicts the statement in the abstract”
Thank you for the remark about the need for clarification, this information has been removed from the abstract (Page 1, Line 29-30).
Comment 3:
“The introduction implies that the sharp rise in thyroid cancer is due to improved imaging and biopsy only. These are not the only causes of increase in thyroid cancer. There are likely multiple other factors involved. For example, more detailed pathology techniques may account for many of the small cancers identified.”
We are grateful for the review of our manuscript. We agree with the statement that causes of increase in thyroid cancer quoted in our research are not the only ones. However, factors mentioned in the introduction seem to be the most significant based on the cited studies.

Reviewer 2 Report
There is currently no consensus regarding the impact of sex on PTC and the authors explore the relevance of sex as a important factor in thyroid cancer, being important for the clinicians,
Despite the authors propose the male sex not be included into the ATA stratification system as a risk factor for persistence/recurrence. However, the findings in the association between male sex and the response of initial therapy in the patients are important and as suggest in the conclusions of the paper these patients may require different treatments strategies
In order to confirm these results, the analysis should be reproduce in a new cohort of patients
Minor points
1.- The characteristics of the study group, Table 1, could be included in the experimental section. (Could be consider as a suggestion)
Author Response
Comment 1: “There is currently no consensus regarding the impact of sex on PTC and the authors explore the relevance of sex as a important factor in thyroid cancer, being important for the clinicians, despite the authors propose the male sex not be included into the ATA stratification system as a risk factor for persistence/recurrence. However, the findings in the association between male sex and the response of initial therapy in the patients are important and as suggest in the conclusions of the paper these patients may require different treatments strategies. In order to confirm these results, the analysis should be reproduce in a new cohort of patients”
We are grateful for the review of our manuscript. Further studies on larger groups of patients should be carried out to assess whether males require treatment modalities or oncological follow-up protocols that are different from those of females as our results in this matter are inconclusive. This clarification has been added in the discussion (Page 11, Line 339-342) and as a conclusion of our study (Page 11, Line 350-351).
Comment 2: “Minor points 1.- The characteristics of the study group, Table 1, could be included in the experimental section. (Could be consider as a suggestion).”
Thank you for this suggestion, we are grateful for the review of our manuscript. We ask the editor to make a decision, in our opinion it is better not to include Table 1 in the experimental section.

This manuscript is a resubmission of an earlier submission. The following is a list of the peer review reports and author responses from that submission.
Round 1
Reviewer 1 Report
This is an interesting study that reported the prognostic value of sex in thyroid cancer. However, the patients included in this study are widely distributed, some of them had radioactive iodine ablation, some of them had a risk factors, but the others did not. Authors need to analyze them separately according to known prognostic systems.
Author Response
Reviewer #1
Comment 1:
“This is an interesting study that reported the prognostic value of sex in thyroid cancer. However, the patients included in this study are widely distributed, some of them had radioactive iodine ablation, some of them had a risk factors, but the others did not. Authors need to analyze them separately according to known prognostic systems.”
Thank you for your comment. All patients with postoperative primary tumor stage greater than pT1aN0-xM0 were eligible for 131-I treatment, followed by a suppressive dose of levothyroxine. Standard procedure of 1100-3700 MBq 131-I was administered, adjusted for TNM status. All details of the 131-I treatment protocol, as well as tests and procedures performed during the evaluation of the effectiveness of primary treatment in patients treated with 131-I in our center, have been described previously. [Kowalska A, Walczyk A, Pałyga I, Gąsior-Perczak D, Gadawska-Juszczyk K, Szymonek M, et al.The treatment effects were assessed 12 months after the administration of 131-I. The Delayed Risk Stratification System in the Risk of Differentiated Thyroid Cancer Recurrence. PLOS ONE. 2016 Apr 14;11(4):e0153242.]
The efficacy of surgical treatment in patients with pT1aN0-xM0, not treated with 131-I was assessed by clinical examination, neck ultrasound, Tg and TgAb levels. Four to 6 weeks after the surgery, before the administration of levothyroxine, scintigraphy of the neck and the whole body scan were performed in patients after total thyroidectomy. If the results indicated that the primary surgery was not effective, the patients were referred for radical surgical treatment. Subsequent follow up were carried out every 6 - 12 months, depending on the degree of poor clinical course, as previously described. [Gąsior-Perczak D, Pałyga I, Szymonek M, Kowalik A, Walczyk A, Kopczyński J, et al. Delayed risk stratification system in pT1aN0/Nx DTC patients treated without radioactive iodine. Endocr Connect. 2017 Oct;6(7):522–7.] The management was standardized depending on the stage of the disease and was independent of the gender.
Reviewer 2 Report
Review: Is male sex a prognostic factor in differentiated thyroid cancer?
The authors present a single center study with 1466 patients to analyze whether male sex has any impact of the prognosis of differentiated thyroid cancer. This is a good conducted study with extensive analysis.
Some concerns are listed below.
Introduction:
page 2; line 46:
you wrote that PTC continues to rise worldwide. Please complete your statement with arguments “why” for example: better resolution of the ultrasound machine, more detected microcarcinomas, etc.
page 2; line 54: …Men tend to be diagnosed at an older age… which age?? Please clarify.
page 2; line 80: in which cases you performed a “lobectomy”? Did you remove the other lobe after getting the histopathological result of “thyroid cancer”? Please clarify.
Results:
Table 5: please rewrite the table to “Effect of clinicopathological factors on worse response to initial therapy” In this way the table is better understandable and is in line with your other findings.
page 4; 3.2.1. - line 124: “…too little time had passed since surgery”. How long is “little time”?
page 5; 3.3. Baseline characteristics according to sex: you wrote that men had a higher risk at an older age? Did you find a cut off?
Discussion
The discussion is weakly argued!! In my opinion this part has to be rewritten!
Your wrote that men had a higher risk of structural disease recurrence than women. In table 6 you presented “no significant difference” between women/men. Please clarify.
I miss completely an meticulous discussion about the reason that men had worse clinicopathological features, worse response to therapy, etc but these factors showed no influence on the cancer related death! So what means “worse”? What is the consequence?
This brings me to my next thoughts?: Why did you conclude, that further studies are required to ascertain whether men should be treated differently from women – if the outcome of recurrence as well as of survival is equal??????
Limitations:
The main structural limitation of the study is the short follow up period of 5 years.
Author Response
Reviewer #2
Comment 1: “Page 2; line 46: You wrote that PTC continues to rise worldwide. Please complete your statement with arguments “why” for example: better resolution of the ultrasound machine, more detected microcarcinomas, etc. ”
We are grateful for the review of our manuscript. Increase in incidence of TC applies mostly to small cancers and it is due to development of widely available and sensitive detection methods such as thyroid ultrasound and the increased availability of ultrasound-guided fine-needle aspiration. Furthermore we observe an increase of larger cancers due to radiation exposure, low iodine intake, genetics, obesity, and smoking. This clarification has been added in lines 46-50.
Comment 2: “Page 2; line 54: Men tend to be diagnosed at an older age… which age?? Please clarify.”
Thank you for your commentary. With the peak at 60-69 years, the rate of papillary thyroid cancer among women is nearly three-times higher than men. In females, the age-specific incidence rate rises sharply at the beginning of the reproductive years, with the increase peaking at 40–49 years, while in men the peak is at 60–69 years. The incidence rates equalize by 85 years of age. Thank you for the remark about the need for clarification, this information has been added in the study (lines 54-58).
Comment 3: “Page 2; line 80: In which cases you performed a “lobectomy”? Did you remove the other lobe after getting the histopathological result of “thyroid cancer”? Please clarify.”
Thank you for your remark concerning performed lobectomy. In Holycross Cancer Center patients are treated with thyroid lobectomy or total thyroidectomy. Lobectomy is performed when the FNAC (fine needle aspiration cytology) results are within the category I, II, III, IV or V, according to the Bethesda system and there is no evidence of nodular changes in the second thyroid lobe. Total thyroidectomy is recommended for patients with Bethesda VI category or for patients with bilateral, multinodular goiter, regardless of FNAC results.
If the histopathology report confirms T1a lesion according to the 8th edition of the AJCC/TNM staging system of thyroid cancer, the lobectomy is considered oncologically sufficient. However, patients presenting more advanced stages of disease need reoperation to remove remnant lobe.
The range of lymph nodes dissection depends on ultrasound assessment and clinical evaluation. FNAC verification should be considered for all cervical nodes with suspicious features, suggestive of presence of metastasis. In all of FNAC - proven thyroid cancer cases the entirety of central compartment lymph nodes is removed if possible.
However, lateral compartment dissection is recommended only in patients with confirmed lymph node metastasis.
Lymph nodes which were not removed during surgery should be reassessed by clinical examination. According to the 8th edition of the AJCC/TNM staging system, patients without lymph node metastases should be classified as N0b.
An appropriate commentary was included into the manuscript in lines 75 -78: “All patients included in the study underwent primary surgical treatment. The scope of surgery lobectomy, total thyroidectomy or total thyroidectomy with central compartment lymphadenectomy. The extent of the surgery and surgical treatment procedures have been described by us previously” [Gąsior-Perczak D, Pałyga I, Szymonek M, Kowalik A, Walczyk A, Kopczyński J, Lizis-Kolus K, Trybek T, Mikina E, Szyska-Skrobot D, Gadawska-Juszczyk K, Hurej S, Szczodry A, Słuszniak A, Słuszniak J, Mężyk R, Góźdź S, Kowalska A. The impact of BMI on clinical progress, response to treatment, and disease course in patients with differentiated thyroid cancer. PLoS One. 2018 Oct 1;13(10):e0204668. doi: 10.1371/journal.pone.0204668. PMID: 30273371; PMCID: PMC6166948.]
The fragment of the aforementioned work:
All patients included in the study were subjected to primary surgical treatment. The scope of surgery included lobectomy, total thyroidectomy (TT), or total thyroidectomy with central compartment lymphadenectomy. In our center, total thyroidectomy with central compartment lymphadenectomy was performed if the primary tumors were >10 mm, multiple or bilateral, or extrathyroidal, or when metastases to the lymph nodes (LN) of the central neck compartment were detected during pre-operative evaluation or surgery. We routinely performed central compartment node dissection on the primary tumor side. On the other hand, we performed bilateral central compartment node dissection when the tumor was bilateral or the LNs were enlarged on the opposite side, as demonstrated during pre-operative staging or surgery. However, the decision to remove lateral LNs depended on the pre- or intra-operative diagnosis of metastases to LNs, or a strong clinical suspicion of their involvement. Lobectomy (total excision of the entire thyroid lobe with isthmus) was performed in patients diagnosed with pre-operative unifocal PTC with a diameter of ≤10 mm, in clinical stage N0b (no lymph node metastases diagnosed in preoperative ultrasound), when there were no evident indications for bilateral surgery in the form of changes visible in the ultrasound in the contralateral lobe, and in patients who had previously undergone lobectomy and were diagnosed with low-risk thyroid cancer. TT without central compartment node dissection was performed in patients with nodular goiter who were diagnosed with PTC ≤10 mm (pT1a) after surgery if they had no evidence of cervical node metastasis in clinical N0b (no LN metastases in postoperative ultrasound) and no distant metastases, and after a careful histopathological examination of the postoperative material to exclude multifocal growth.
Comment 4: “Table 5: please rewrite the table to “Effect of clinicopathological factors on worse response to initial therapy” In this way the table is better understandable and is in line with your other findings.”
Thank you for your remark, the caption above the table has been changed (line 231).
Comment 6: “Page 4; 3.2.1. - line 124: “…too little time had passed since surgery”. How long is “little time”?”
Thank you for your comment. We deleted the imprecise statement and replaced it in lines 139-141 with the following one: It was not possible to assess treatment response in 123 patients (8.4%) because their follow up period was shorter than 12 months (primary treatment response is assessed after at least 12 months). [Haugen BR, Alexander EK, Bible KC, Doherty GM, Mandel SJ, Nikiforov YE et al. 2015. American Thyroid Association Management Guidelines for Adult Patients with Thyroid Nodules and Differentiated Thyroid Cancer: The American Thyroid Association Guidelines Task Force on Thyroid Nodules and Differentiated Thyroid Cancer. Thyroid 2016, 26(1):1–133.]
Comment 7: “Page 5; 3.3. Baseline characteristics according to sex: you wrote that men had a higher risk at an older age? Did you find a cut off?”
According to the updated AJCC TNM Staging System for Differentiated and Anaplastic Thyroid Cancer (8th Edition) the cut off age that worsens the prognosis and influence staging is 55.
Comment 8: “You wrote that men had a higher risk of structural disease recurrence than women. In table 6 you presented “no significant difference” between women/men. Please clarify.”
Thank you for your remark, discussion has been revised according to the new outcomes of multivariable analysis (lines 366-369).
Comment 9: “I miss completely an meticulous discussion about the reason that men had worse clinicopathological features, worse response to therapy, etc but these factors showed no influence on the cancer related death! So what means “worse”? What is the consequence? This brings me to my next thoughts?: Why did you conclude, that further studies are required to ascertain whether men should be treated differently from women – if the outcome of recurrence as well as of survival is equal?????”
We are grateful for the review of our manuscript, discussion has been changed according to the results of conducted multivariable analysis (lines 390-396).
Comment 10: “The main structural limitation of the study is the short follow up period of 5 years.”
Thank you for the remark concerning follow-up. We agree with the Reviewer’s suggestions. The longer follow-up probably delivers additional information about recurrence and death, because other clinicopathological features are not associated with follow-up (response to therapy). However, the highest risk of recurrence is during the first 5 years of follow-up. Late recurrence is possible even after many years after primary treatment. [Durante, C et.al. "Papillary thyroid cancer: time course of recurrences during postsurgery surveillance." J Clin Endocrinol Metab. 2013 Feb;98(2):636-42)]
Reviewer 3 Report
The study aims to analyze the differences in clinicopathological features, responses to treatment and outcomes between women and men with differentiated thyroid cancer using the results from a single institution. It is generally well written and comments on many of the aspects of the disease that influence outcomes. Although it is commonly known that men have worse features this study does add to the literature in that it analyzes the outcomes according to the ATA criteria of risk. However, there are several areas and details which have not been included which would affect interpretation of the data and I have laid these out in the specific questions and comments below. 1. The grammar in some parts of the manuscript needs correction e.g. in the first sentence of the abstract “whether male sex is one of unfavorable risk factors” and line 56 “The purpose of present study”. 2. It seems that poorly differentiated thyroid cancer was included in the study, whereas the title, introduction and methods indicated that the study is only for differentiated thyroid cancer..this should preclude poorly differentiated thyroid cancer. Otherwise, the manuscript can be amended to indicate that well and poorly differentiated thyroid cancer were studied. 3. The incidence of lymph node metastasis is very low compared with most data (with many studies indicating up ~ 60 % central lymph node metastasis). There is no information about the details of the procedures..whether they included central (and/or lateral) lymph node dissection. Were lymph nodes sampled and frozen section utilized? If no central neck dissections were performed, were these lymph node metastasis found on final pathology due to incidental removal of lymph nodes? Were this micrometastases? Was preoperative cytological assessment of lymph nodes utilized? The answer to these questions affects the interpretation of the findings. 4. The authors comment that many factors affect recurrence, and sex is not such a factor. However, the findings in their study (and of others) indicate that these diverse poor prognostic factors are more common in men than women. Could the authors comment on this and their thoughts about why more recurrence is not seen in men despite the worse clinicopathological features. Is the follow up surveillance techniques used sufficient to detect recurrence? It is not clear, for example, how many patients had lobectomy versus total thyroidectomy and which of the latter group had RAI. The ability to detect recurrence will be affected by these factors and the proficiency of those performing surveillance scans. Also, it is confusing that in the results section it is stated that there was no difference in recurrence between the sexes; whereas in the discussion a difference in structural recurrence is first noted, then it is stated that sex does not influence recurrence. 5. The results for the various cancers are grouped together, however, it is likely that the different cancer types (and subtypes) will have different outcomes and responses to treatment. It would be useful to analyze the data using the different cancer types to assess the true variance between sexes within each cancer type/subtype.Author Response
Reviewer #3
Comment 1: “The study aims to analyze the differences in clinicopathological features, responses to treatment and outcomes between women and men with differentiated thyroid cancer using the results from a single institution. It is generally well written and comments on many of the aspects of the disease that influence outcomes. Although it is commonly known that men have worse features this study does add to the literature in that it analyzes the outcomes according to the ATA criteria of risk. However, there are several areas and details which have not been included which would affect interpretation of the data and I have laid these out in the specific questions and comments below.
The grammar in some parts of the manuscript needs correction e.g. in the first sentence of the abstract “whether male sex is one of unfavorable risk factors” and line 56 “The purpose of present study”.
Thank you for your comment, the work has been thoroughly checked for mistakes. Grammar was revised as requested. In the first sentence of the abstract (lines 24-26) phrase “The purpose of the present study was to find out whether male sex is one of unfavorable risk factors that have a negative effect on the course of differentiated thyroid cancer” was changed to “Identification of unfavorable risk factors that have a negative effect on the course of differentiated thyroid cancer (DTC) is a clinical problem. The purpose of the present study was to find out whether male sex is one of these factors.” Lines 59-62 were revised to “The aim of present study was to analyze influence of male sex on more frequent occurrence of DTC with unfavorable clinicopathological features, and to examine whether men show a different response to therapy.” . However if you still find it necessary to further improve the manuscript as far as the spelling is concerned, we are ready to send it again to a proofreading service
Comment 2: “It seems that poorly differentiated thyroid cancer was included in the study, whereas the title, introduction and methods indicated that the study is only for differentiated thyroid cancer..this should preclude poorly differentiated thyroid cancer. Otherwise, the manuscript can be amended to indicate that well and poorly differentiated thyroid cancer were studied”
Thank you for your insight regarding the need to clarify whether poorly differentiated thyroid cancer was included in the study. In our study, we included all differentiated thyroid cancers: well-differentiated (PTC, FTC, Hürthle cell carcinomas) and poorly differentiated thyroid cancers and included this information in the title of the work. As the clinical courses of poorly differentiated carcinomas, follicular carcinomas and Hürthle cell carcinomas are worse than PTC, we corrected our study by dividing the studied patients into PTC and non-PTC groups (due to the small size of the groups with poorly differentiated PTC, FTC and Hürthle cell carcinoma, we did not analyze them separately). We would not like to remove poorly differentiated TC from the study, as one of the more important findings is that men are more likely to have non-papillary TC, therefore, when diagnosing thyroid tumor in men, these types of TC should be considered.
Comment 3: “The incidence of lymph node metastasis is very low compared with most data (with many studies indicating up ~ 60 % central lymph node metastasis). There is no information about the details of the procedures..whether they included central (and/or lateral) lymph node dissection. Were lymph nodes sampled and frozen section utilized? If no central neck dissections were performed, were these lymph node metastasis found on final pathology due to incidental removal of lymph nodes? Were this micrometastases? Was preoperative cytological assessment of lymph nodes utilized? The answer to these questions affects the interpretation of the findings.”
Thank you for your remark about the lack of procedures description. The description of the scope of the thyroid and cervical lymph node surgery performed in our center has been added to the experimental section (Lines 75-78). The presence of micrometastases to cervical lymph nodes does not worsen the prognosis and extensive nodal surgery contributes to a significant increase in surgical complications in the form of hypoparathyroidism. According to the ATA recommendations, if no cervical lymph node metastases are found in the histopathological examination, we perform a postoperative clinical assessment of the state of the perform lymph nodes and imaging examinations (US), and if no suspicious nodal metastases are found, we classify them as N0b and low-risk according to ATA. [Haugen BR, Alexander EK, Bible KC, Doherty GM, Mandel SJ, Nikiforov YE et al. 2015. American Thyroid Association Management Guidelines for Adult Patients with Thyroid Nodules and Differentiated Thyroid Cancer: The American Thyroid Association Guidelines Task Force on Thyroid Nodules and Differentiated Thyroid Cancer. Thyroid 2016, 26(1):1–133.]
Comment 4: ”The authors comment that many factors affect recurrence, and sex is not such a factor. However, the findings in their study (and of others) indicate that these diverse poor prognostic factors are more common in men than women. Could the authors comment on this and their thoughts about why more recurrence is not seen in men despite the worse clinicopathological features. Is the follow up surveillance techniques used sufficient to detect recurrence? It is not clear, for example, how many patients had lobectomy versus total thyroidectomy and which of the latter group had RAI. The ability to detect recurrence will be affected by these factors and the proficiency of those performing surveillance scans. Also, it is confusing that in the results section it is stated that there was no difference in recurrence between the sexes; whereas in the discussion a difference in structural recurrence is first noted, then it is stated that sex does not influence recurrence.”
Well known risk factors generally relate to worse clinical outcomes of the disease, not selectively to the recurrence. ATA risk stratification system selects features which could assess the risk of worse clinical course - that is, disease persistent after primary treatment and the recurrence. In our study we define recurrence as a relapse after a NED period. It occurs rarely - only 10 patients have demonstrated it, therefore it is difficult to indicate statistic relevance of the effect of various factors in such a small group. However, in our study, the male patients tend to often present more severe clinicopathological features which leads to worse response to the initial treatment (non-excellent response). Discrepancies in the definition of recurrence are the main cause of inconsistencies in observation results, e.g. in one of the multicenter studies: [Zahedi A, et al. Risk for Thyroid Cancer Recurrence Is Higher in Men Than in Women Independent of Disease Stage at Presentation. Thyroid. 2020 Jun;30(6):871-877. doi: 10.1089/thy.2018.0775. Epub 2019 Nov 13. PMID: 31524071.] the authors had different definitions of recurrence and did not specify the meaning of ‘disease free’ term. Therefore, many patients with persistent disease are included into the recurrent disease group and regardless of the fact that the same criteria were applied for both male and female patients, it has an immense influence on the final assessment of disease recurrence risk or persistent disease. Recurrence in our study was assessed as the appearance of a new focal disease, shown in imaging studies and increase in the concentration of TC markers in patients with excellent response to the initial treatment (as defined by ATA) after the NED period.
Lobectomy is considered as an oncologically sufficient procedure only in the treatment of microcarcinomas. More advanced thyroid carcinomas require single stage or two stage thyroidectomy with complementary 131 I treatment, regardless of patient's sex. Due to the fact that the previous analysis did not find an impact of low or high 131 I doses on cancer recurrence, we did not analyze the effect of 131 I activity which we administered during treatment in our study.
[Vardarli et al. Longer-term recurrence rate after low versus high dose radioiodine ablation for differentiated thyroid Cancer in low and intermediate risk patients: a meta-analysis BMC Cancer (2020) 20:55]
Comment 5: “The results for the various cancers are grouped together, however, it is likely that the different cancer types (and subtypes) will have different outcomes and responses to treatment. It would be useful to analyze the data using the different cancer types to assess the true variance between sexes within each cancer type/subtype.”
Thank you for your remark concerning the need for separate analysis using the different thyroid cancer types. Additional analysis by type of cancer - PTC and non-PTC has been added in our study.
Reviewer 4 Report
The authors present a retrospective single center clinico-pathological analysis on DTC with special focus on male sex as a negative prognostic factor for patient outcomes. In my view, the manuscript is of interest to the clinical community and summarizes well valuable information; therefore I believe it would be of interest to the readership of JCM. However, there are some points that need to be addressed with respect to the study methodology/data analysis. The manuscript would benefit from major revision and clarification with respect to its limitations.
Major Comment:
Please perform multivariable logistic regression analyses of to investigate the effect of clinicopathological factors on: 1) excellent response to initial therapy; 2) Recurrence at any time during follow-up.
The Reviewer acknowledges that multiavarable Cox-regression analysis with respect to DSS is not feasible due to the overall favorable prognosis of DTC and the scarcity of DTC-related deaths in this cohort. If the quality of data permits such analysis, then PFS multivariable Cox-regression analysis could be undertaken with regards to survival outcomes.
However, the significance of gender as a prognostic factor should be substantiated in models of multivariable analysis as specified above in respect of response to initial treatment and recurrence during follow-up.
Author Response
Reviewer #4
Comment: “Please perform multivariable logistic regression analyses of to investigate the effect of clinicopathological factors on: 1) excellent response to initial therapy; 2) Recurrence at any time during follow-up.
The Reviewer acknowledges that multivariable Cox-regression analysis with respect to DSS is not feasible due to the overall favorable prognosis of DTC and the scarcity of DTC-related deaths in this cohort. If the quality of data permits such analysis, then PFS multivariable Cox-regression analysis could be undertaken with regards to survival outcomes.
However, the significance of gender as a prognostic factor should be substantiated in models of multivariable analysis as specified above in respect of response to initial treatment and recurrence during follow-up.“
Thank you for your remark, univariable and multivariable logistic regression analyses were performed to investigate the impact of clinicopathological features on non excellent response to initial therapy and no NED at the final follow up. Mentioned analyses have been conducted in our study in table 5.
Multivariable analysis of regression to investigate the impact of clinicopathological features on recurrence could not have been conducted due to the small number of patients with cancer recurrence (2 patients diagnosed with papillary thyroid cancer and 8 patients with non-PTC). Result of such analysis would not be reliable.
Round 2
Reviewer 1 Report
This is an interesting study that investigated the effect of sex in thyroid cancer. However, the effect of male sex in thyroid cancer has been investigated before in the view of prognosis or aggressiveness, is there a novel finding in this study? In addition, there were few patients with thyroid cancer other than PTC, therefore, the characteristics in thyroid cancer other than PTC may not be adequate to be analyzed in this study.Author Response
Comment 1: “This is an interesting study that investigated the effect of sex in thyroid cancer. However, the effect of male sex in thyroid cancer has been investigated before in the view of prognosis or aggressiveness, is there a novel finding in this study”
We are grateful for the review of our manuscript. Our study proves that male sex is associated with more frequent occurrence of PTC with some unfavorable clinicopathological features but this does not contribute to worse response to initial therapy and final status and clinical outcome. A novel finding in our study is an evaluation of the effect of sex on response to the initial treatment. The delayed risk stratification system (DRS) based on response to initial therapy correlated more strongly with final status of disease (NED/non-NED) than the early stratification systems created by the ATA. In our study, sex was not a factor that predicted a worse response to initial therapy, and there was no relationship between sex and final status of disease.
Comment2: “In addition, there were few patients with thyroid cancer other than PTC, therefore, the characteristics in thyroid cancer other than PTC may not be adequate to be analyzed in this study.”
Thank you for your remark. We agree with the reviewer’s suggestion and an analysis of other than papillary thyroid cancer types has been excluded from our study.
Reviewer 2 Report
No further suggestions
Author Response
We are grateful for the review of our manuscript. Grammar was revised as requested by the professional proofreading service.
Reviewer 3 Report
The authors have improved the manuscript significantly and especially by performing a separate analysis of the PTC group versus non-PTC. Further refinement of the grammar is required. Also, the question of cancer recurrence and the explanation for the lack of difference seen between the genders can be better explained.
Author Response
Comment 1: “The authors have improved the manuscript significantly and especially by performing a separate analysis of the PTC group versus non-PTC. Further refinement of the grammar is required. Also, the question of cancer recurrence and the explanation for the lack of difference seen between the genders can be better explained.”
We are grateful for the review of our manuscript. Grammar was revised as requested by the professional proofreading service.
Recurrence defined as a relapse of disease in patients, who achieved an excellent response to therapy after the NED period of PTC is a very rare phenomenon. It was observed only in eight of our 1386 patients in median 5-year follow up. Therefore, it was not possible to assess statistical significance in such a small group.
Reviewer 4 Report
The Authors provide an extensive revision of the manuscript, but fail in my opinion to provide evidence in a comprehensive manner on the role of gender in TC.
- The Authors need to specify the variables entered in the multivariable regression model. For example ”ATA initial risk” viariable along with other clinicopathological parameters in the regression model would introduce multicollinearity.
- Hazard rations should be provided following standard nomenclature and STROBE guidelines. Results should be reported in the abstract of the manuscript accordingly.
- The sample size of non PTC patients does not allow separate mutlivariable analysis. Therefore this part should be omitted. In addition, the results of univariale analysis should interpreted with caution.
- A p-value of 0.25 in multivariable analysis for gender is not in accordance with the conclusions of the study (Table 5). What is the HR of male gender for non-excelend response?
- The number of deaths and recurrences along with the length of patient follow-up in this study precludes any safe conclusion to be made on recurrence and survival estimates. The conclusions of the study as stated in the abstract are not supported by the data analysis.
Author Response
Comment 1: “The Authors provide an extensive revision of the manuscript, but fail in my opinion to provide evidence in a comprehensive manner on the role of gender in TC. The Authors need to specify the variables entered in the multivariable regression model. For example ”ATA initial risk” variable along with other clinicopathological parameters in the regression model would introduce multicollinearity.”
We are grateful for the review of our manuscript. Thank you for your remark about the need of clarification of the variables examined in the multivariate Cox-regression analysis. The “ATA initial risk” variable, along with other clinicopathological parameters , was not entered into the multivariable logistic regression model to avoid introduction of multicollinearity. It has been explained in the lines 203 to 204 and 225 to 227 . Explanations of which variables were entered in the multivariate model have been added below the table 5 and table 7.
Comment 2: “Hazard rations should be provided following standard nomenclature and STROBE guidelines. Results should be reported in the abstract of the manuscript accordingly.”
Thank you for your remark. Manuscript have been inspected again in the terms of following standards of STROBE guidelines. Abstract has been revised as requested and results of performed analyses have been included.
Comment 3: “The sample size of non PTC patients does not allow separate multivariable analysis. Therefore this part should be omitted. In addition, the results of univariate analysis should interpreted with caution.”
Thank you for your remark. An analysis of other than papillary thyroid cancer types has been excluded from our study. We agree with the reviewer’s suggestion that the results of univariate analysis must be carefully interpreted. Therefore conclusions in our study are based on multivariate analysis.
Comment 4: “A p-value of 0.25 in multivariable analysis for gender is not in accordance with the conclusions of the study (Table 5). What is the HR of male gender for non-excellent response?”
In the table 5 in the univariate analysis p-value is 0.0273. However multivariate analysis did not confirm this statistical significance (p=0.2927) what has been stated in the results. Response to treatment is assessed after 12 months from initial treatment. Therefore hazard ratio seem to have no meaning.
Comment 5: “The number of deaths and recurrences along with the length of patient follow-up in this study precludes any safe conclusion to be made on recurrence and survival estimates. The conclusions of the study as stated in the abstract are not supported by the data analysis”
Thank you for your remark concerning the small number of fatal events in our study. In the group of patients diagnosed with PTC there were no TC-related deaths, therefore we could not examine the effect of sex on mortality. Cancer types other than papillary were excluded from the research. Abstract has been revised as suggested.